**REVIEW-SYMPOSIUM**

# Cortico-thalamocortical interactions for learning, memory and decision-making

Brook A. L. Perry[1] , Juan Carlos Mendez[1,2] and Anna S. Mitchell[1]

[1]*Department of Experimental Psychology, University of Oxford, Oxford, UK*
[2]*College of Medicine and Health, University of Exeter, Exeter, UK*

Handling Editors: Ian Forsythe & Richard Carson

The peer review history is available in the Supporting information section of this article (https://doi.org/10.1113/JP282626#support-information-section).

<div style="text-align:center"><em>The Journal of Physiology</em></div>

**Abstract** The thalamus and cortex are interconnected both functionally and anatomically and share a common developmental trajectory. Interactions between the mediodorsal thalamus (MD) and different parts of the prefrontal cortex are essential in cognitive processes, such as learning and adaptive decision-making. Cortico-thalamocortical interactions involving other dorsal thalamic nuclei, including the anterior thalamus and pulvinar, also influence these cognitive processes. Our work, and that of others, indicates a crucial influence of these inter-dependent cortico-thalamocortical neural networks that contributes actively to the processing of information within the cortex. Each of these thalamic nuclei also receives potent subcortical

This symposium review forms part of the '*Decoding Prefrontal Cortical Physiology: Circuits of Cognition*' symposium held at *Physiology 2021* in July 2021, and organized by Professor Matt Jones.

inputs that are likely to provide additional influences on their regulation of cortical activity. Here, we highlight our current neuroscientific research aimed at establishing when cortico-MD thalamocortical neural network communication is vital within the context of a rapid learning and memory discrimination task. We are collecting evidence of MD–prefrontal cortex neural network communication in awake, behaving male rhesus macaques. Given the prevailing evidence, further studies are needed to identify both broad and specific mechanisms that govern how the MD, anterior thalamus and pulvinar cortico-thalamocortical interactions support learning, memory and decision-making. Current evidence shows that the MD (and the anterior thalamus) are crucial for frontotemporal communication, and the pulvinar is crucial for frontoparietal communication. Such work is crucial to advance our understanding of the neuroanatomical and physiological bases of these brain functions in humans. In turn, this might offer avenues to develop effective treatment strategies to improve the cognitive deficits often observed in many debilitating neurological disorders and diseases and in neurodegeneration.

(Received 15 December 2021; accepted after revision 30 June 2022; first published online 19 July 2022)

**Corresponding author** Anna S. Mitchell: Department of Psychology, Speech and Hearing, University of Canterbury, Private Bag 4800, Christchurch 8140, New Zealand. Email: anna.mitchell@canterbury.ac.nz

**Abstract figure legend** Overlapping prefrontal cortex connectivity for the subdivisions of the mediodorsal thalamus [MD: magnocellular MD (MDmc), parvocellular MD (MDpc) and lateral MD (MDl)], the anterior thalamus (ATN) and the pulvinar.

## Background

For >120 years, clinical evidence has highlighted that damage to, or dysfunction of, the mediodorsal thalamus (MD), the prefrontal cortex (PFC) and/or the interconnections between these two structures is associated with cognitive deficits, particularly in the domains of learning and memory (Kopelman, 2015; Markowitsch, 1982; Pergola et al., 2018; Victor et al., 1971). Likewise, neuropathology and neuroimaging reports have associated changes in MD–PFC white matter connectivity with the cognitive impairments found in several neurological and psychiatric disorders, including Wernicke–Korsakoff syndrome and neurodegeneration related to alcoholism (Kopelman, 2015; Segobin et al., 2019), thalamic stroke (Danet et al., 2015; Pergola et al., 2018), affective disorders (Pergola et al., 2018), frontotemporal dementia (Bocchetta et al., 2020) and Parkinson's disease (Harrington et al., 2020). Furthermore, neurodevelopmental disorders, particularly schizophrenia (Parnaudeau et al., 2018; Pergola et al., 2015), also involve changes in MD–PFC thalamocortical interactions which, along with other brain changes, lead to the observed cognitive deficits.

This clinical evidence prompted earlier neuroscientific studies involving animal models that reaffirmed the crucial influence of the MD in cognition (e.g. Aggleton & Mishkin, 1983a, b; Gaffan & Parker, 2000; Isseroff et al., 1982; Mitchell, 2015; Mitchell & Dalrymple-Alford, 2005; Mitchell & Chakraborty, 2013). More recently, studies using animal models have begun to capture the essence of the contributions of the thalamus and cortex as they work in partnership during learning, memory, decision-making and other higher cognitive processes (Jones, 2009). For

**Brook A. L. Perry** joined the Thalamus, Cortex and Cognition Lab in 2018 as a postdoctoral researcher leading the neurophysiological recordings and analyses in non-human primates. **Juan Carlos Mendez** joined the Thalamus, Cortex and Cognition Lab in 2020 as a postdoctoral researcher to support the neurophysiological recording analyses and complete neuroimaging studies in non-human primates. **Anna S. Mitchell** joined the Department of Experimental Psychology at Oxford University in 2004 as a postdoctoral researcher under the tutelage of Professor David Gaffan. She established the Thalamus, Cortex and Cognition Lab in 2009 with funding from the Medical Research Council Career Development Award. She is now a Wellcome Trust Senior Research Fellow in Basic Biomedical Sciences and an Associate Professor. Her group studies the neurobiology of learning, memory and decision-making, with a focus on the contributions of the dorsal medial thalamus and interconnected structures in health and disease, using animal models and humans.

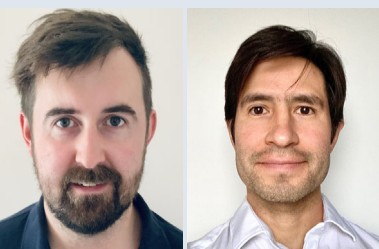
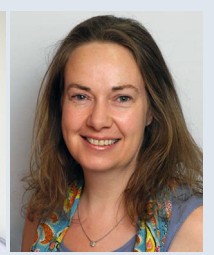

example, non-human primates (NHPs) with permanent neurotoxic damage to the MD in one hemisphere and a contralateral PFC ablation were shown to have extensive deficits during rapid visuospatial discrimination learning and adaptive decision-making tasks (Browning et al., 2015; Izquierdo & Murray, 2010). Likewise, in mice, temporary disruption of the MD or medial PFC using optogenetics during an attentional control task caused increased errors, especially when task parameters became more complex (Mukherjee et al., 2021). In rats, chemogenetic inactivation of MD → PFC and PFC → MD pathways using a dual viral vector strategy captured the influence of these pathways during an instrumental learning task (Alcaraz et al., 2018). However, only the inactivation of the MD → PFC pathway caused deficits in evaluating the causal relationships between actions and outcomes. In contrast, inactivation of either pathway impaired the re-evaluation of changing rewards. Finally, recent neuroimaging studies in healthy volunteers support the involvement of thalamocortical interactions involving the MD or the pulvinar and of additional thalamic structures in cognitive processes (e.g. Hwang et al., 2021; Kosciessa et al., 2021).

More specifically, in the case of the pulvinar, neuroscientific studies involving NHPs have established its crucial influence over interconnected cortical areas in visuospatial attentional processes (e.g. Fiebelkorn et al., 2019; Saalmann et al., 2012; for review, see Fiebelkorn & Kastner, 2020), with a recent computational model of cortico-pulvinar-cortical interactions suggesting how this feedback and feedforward neural circuitry might influence higher cognitive processes (Jaramillo et al., 2019). Jaramillo et al. (2019) showed that changes in excitability of the pulvinar caused influential changes in the computational priorities of their model, leading to changes in behavioural outcomes. Clinical studies have also identified the contribution of the pulvinar to visuospatial whereabouts, orientation and movement (Snow et al., 2009; Wilke et al., 2018). Additionally, functional and diffusion imaging analyses has identified distinct differences in thalamocortical connectivity of the human pulvinar with the dorsal pulvinar interconnected to the frontal and parietal cortex and the ventral pulvinar interconnected to the visual cortex (Arcaro et al., 2015). An interesting recent review proposes the pulvinar as a hub for multisensory integration, given its extensive cortico-thalamocortical connectivity (Froesel et al., 2021). Clearly, complementary evidence from humans and animal models is essential for the development of a direct link that supports further testing and assessments in neuropsychology and psychiatry (Pergola et al., 2018; Scott & Bourne, 2022).

Neuroscience endeavours ultimately to understand the workings of the human brain. Consequently, further understanding of the influence of the MD, the pulvinar

and the anterior thalamic (ATN) nuclei in cognitive processes has the potential to lead to effective treatment targets for many clinical populations. Although human participants have the advantage of being easier to train to complete many complex cognitive tasks, access to intracranial neural recordings is very limited. Combining such recordings with invasive brain perturbations is not possible. Therefore, it is still crucially important to use animal models to establish when and how cortico-thalamocortical interactions are necessary during many higher cognitive functions. To this end, we need to pinpoint when thalamocortical interactions are important for successful cognitive processes to occur. The aforementioned evidence from animal models suggests that MD–PFC interactions are particularly important during associative learning of newly processed sensory information coming from association cortex areas, when several threads of task-relevant information need to be bound together rapidly or as task demands increase (requiring a need for updating of associative links) during a testing session (Chakraborty et al., 2016; Mitchell, 2015; Mukherjee et al., 2021; Schmitt et al., 2017), although further studies are needed to support these proposals fully.

## Why might dorsal thalamic nuclei be viable targets for treatment in higher cognitive processes?

The nuclei of the dorsal thalamus have the advantage of being key nodal structures, with each one interconnected to discrete, interdependent cortico-thalamocortical neural networks. They are themselves small enough to be targeted in their entirety with viral vectors coding for opsins or designer receptors and might then have the potential to be manipulated selectively (Alcaraz et al., 2018; Barnett et al., 2021; Courtiol et al., 2019; Mukherjee et al., 2021; Schmitt et al., 2017).

Very recently, optogenetic theta burst stimulation of glutamatergic neurons in the ATN restored the ability of rats with permanent mammillothalamic tract lesions to perform a spatial working memory task (Barnett et al, 2021). Furthermore, ATN stimulation after mammillothalamic tract lesions enhanced rhythmic electrical activity and increased immediate early gene expression across memory-related brain regions, suggestive of wide-ranging and broad effects of thalamic modulation (Barnett et al, 2021). The use of two viral vectors to transfect, and subsequently manipulate, both the soma and the terminals of neurons or glial cells was first developed in NHPs (Kinoshita et al., 2012). In time, with further translational cross-species advances in the development of effective viral vectors in primates, it seems plausible to envisage this or similar viral vector approaches being used effectively to target structures such as the subdivisions of the MD, the ATN or the

pulvinar in humans based on their differences in neuro-chemical parcellations. A similar proposal has already been suggested for dorsal thalamic nuclei (e.g. Barnett et al., 2018; Courtiol et al., 2019). However, additional translational work in NHPs is essential to develop this technology further to target these deep brain structures without causing extensive damage to overlying structures and to ensure robust thalamic receptor uptake of the viruses.

Subdivisions in the MD, the ATN and the pulvinar are proposed to be higher-order thalamic nuclei (Guillery, 1995; Halassa & Kastner, 2017; Mitchell, 2015; Perry & Mitchell, 2019; Schwartz et al., 1991), which are characterized by having their primary driving inputs coming from layer V pyramidal neurons in the cortex, rather than from peripheral or sensory structures, as is the case for first-order thalamic nuclei (Guillery, 1995; Halassa & Sherman, 2019; Perry et al., 2021; Sherman & Guillery, 2013). This alternative view of cortico-thalamocortical interactions 'leads to the understanding that the thalamus continues to contribute to the processing of information within cortical hierarchies' (Sherman, 2016) rather than being a passive relay of sensory information. Although subdivisions of the MD, the pulvinar and the ATN are proposed to be higher-order thalamic nuclei, it is important to note that each of these nuclei also receives potent subcortical driving inputs (Sherman, 2016) that are also likely to contribute to their respective influences within cortical information-processing hierarchies. In particular, for the magnocellular subdivision of the MD (MDmc), recent electron microscopic work indicates that at least some of these subcortical driving inputs are coming from the amygdala (Timbie et al., 2020), confirming earlier tracer work in NHPs (Aggleton & Mishkin, 1984; Russchen et al., 1987). Given the computational modelling proposal of Jaramillo et al. (2019), it seems highly likely that these additional cortical, subcortical and neuromodulatory inputs to higher-order thalamic structures, such as the MD, the ATN and the pulvinar, are helping to change their levels of excitability, leading to differential downstream influences on behaviour.

## A long road to identifying the MD influence on higher cognitive processes

We are still far from a complete understanding of how each thalamic nucleus contributes to the overall neural network. For example, deep brain stimulation of the subthalamic nucleus, the globus pallidus or the thalamus, which form part of an extended neural circuit, are all effective current treatments in patients with differing symptoms in Parkinson's disease and other motor impairments (Lozano et al., 2002). Nevertheless,

it is widely recognized that we do not yet understand fully the underlying mechanisms that are supporting these effective stimulation protocols.

Using lesion studies in animal models, neuroscientists were initially intent on dissecting the contribution of individual thalamic nuclei to higher cognitive processes, because damage to these nuclei has been determined to lead to extensive cognitive deficits (e.g. Harding et al., 2000; Kril & Harper, 2012; Markowitsch, 1982; Victor et al., 1971). However, the underlying systemic dysfunction stemming from the thalamic injury can impact on multiple brain regions and the central and peripheral nervous systems. In the case of patients with thalamic strokes, damage is very rarely focal and, instead, typically involves multiple thalamic nuclei and adjacent white matter tracts (Danet et al., 2015; Pergola et al., 2018).

Nevertheless, experimental neurotoxic lesion studies have allowed us to delineate some of the contributions of individual primate MD subdivisions or the pulvinar to higher cognitive processes. Causal neurotoxic lesion studies have identified the crucial contribution of the MDmc during rapid learning of new visuospatial discriminations (Mitchell, Baxter et al., 2007) and during adaptive value-based or probabilistic decision-making tasks (Chakraborty et al., 2016; Mitchell, Browning et al., 2007). However, MDmc damage does not impair retention of visuospatial discriminations (Mitchell & Gaffan, 2008) or preoperatively acquired strategies for solving decision-making tasks (Mitchell, Baxter et al., 2007). In contrast, the adjacent parvocellular subdivision of the MD (MDpc) is not involved in learning new visuospatial discriminations (Chakraborty et al., 2019). Taken together, this and other evidence from rodent studies has been very useful in developing our understanding of the particular higher cognitive processes that are disrupted after selective MD perturbations combined with changes in the interconnected frontal cortex (Browning et al., 2015; Izquierdo & Murray, 2010). Likewise, many causal lesion or inactivation studies of monkey pulvinar have highlighted its crucial influence on visuospatial attention, cognition and sensorimotor processes (e.g. Bridge et al., 2016; Purushothaman et al., 2012; Wilke et al., 2010, 2013; Zhou et al., 2016). However, as with any one method, lesion studies provide insight about how the rest of the brain functions in the absence of a particular structure, but are unable to indicate when or how the structures themselves are contributing to these processes.

Neuroimaging is another method that can provide an understanding of which neural networks are changed as a consequence of engaging in higher cognitive processes. For example, researchers in our laboratory have performed neuroimaging analyses of brain changes as a consequence of learning the visuospatial discrimination learning task mentioned in the above lesion studies.

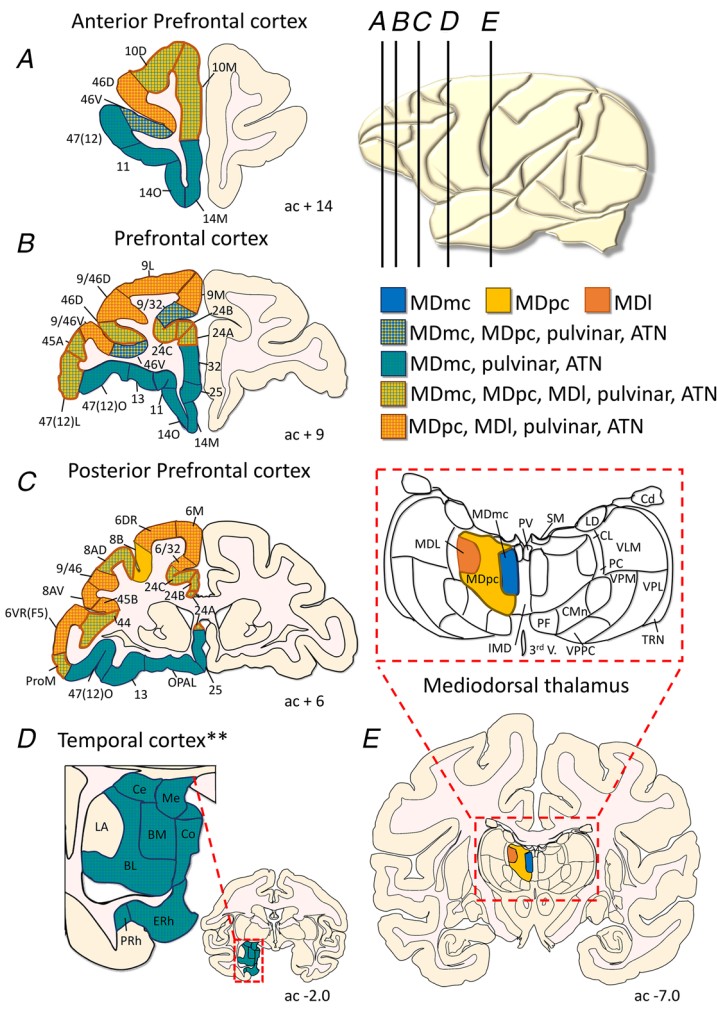

**Figure 1. Cortico-thalamocortical connections**
Schematic representations of the main cortical and subcortical connections of the three mediodorsal thalamus (MD) subdivisions [magnocellular MD (MDmc), parvocellular MD (MDpc) and lateral MD (MDl)] in macaque coronal plates. Overlapping frontal and temporal cortex connections to and from the pulvinar are highlighted, but not all neuroanatomical connectivity of the pulvinar is shown. Overlapping frontal cortex connections of the anterior thalamus (ATN) are shown (for complete details, please refer to Perry et al., 2021). The anterior–posterior position of each coronal section is given relative to the anterior commissure (ac) based on Paxinos, Huang and Toga (2000). The relative position of these coronal plates in the macaque brain is indicated on the sagittal plane in the top right illustration (*A–E*). The anatomical connectivity of the MD is based on the work of several laboratories (e.g. Aggleton & Mishkin, 1984; Russchen et al., 1987; Saunders et al., 2005; Schwartz et al., 1991; Timbie et al., 2020). It is apparent across species, but especially in the macaque, that the MDmc forms part of a distinct frontotemporal circuit receiving inputs from the perirhinal (PRh) and entorhinal (ERh) cortex and the amygdala (basomedial [BL], basomedial [BM], corticomedial [Co], medial [Me] and centromedial [Ce]), in addition to more ventral and ventro-medial regions of prefrontal (areas 25, 32 and orbital periallocortex [OPAL]) and orbitofrontal cortex [areas 11, 13, 14 and 47(12)]. In contrast, the MDpc and MDl subdivisions tend to interact with more dorsolateral frontal regions (areas 9, 45 and 46). Interestingly, in the macaque, the anterior cingulate cortex (areas 24A, B and C) and the frontal pole (areas 10D and 10M) appear to be convergence points for connections with all three MD subdivisions, perhaps indicating a special role for these regions in integrating thalamocortical and corticocortical information. NB The pulvinar sends inputs to the amygdala and temporal cortex structures and has reciprocal connectivity with layer VI of the frontal cortex (e.g. Elorette et al., 2018; Jones & Burton, 1976; Rafal et al., 2015; Shipp, 2003). **The ATN does not connect directly to the amygdala or the perirhinal cortex. However, it is interconnected to the entorhinal cortex (refer to Perry et al., 2021).

Our results indicated that resting-state functional MRI and structural connectivity changes occurred between the dorsal medial thalamic nuclei and interconnected cortical regions (Pelekanos et al., 2020). Interestingly, however, not only were changes noted between the thalamus and frontal cortex; in addition, we found changes in structural and functional connectivity between structures in the temporal cortex and the dorsal medial thalamus and between the temporal cortex and the ventrolateral PFC (Pelekanos et al., 2020). In this context, it is important to mention that the perirhinal cortex, in the temporal cortex, projects to the MDmc (Russchen et al., 1987; Saunders et al., 2005). Other studies in monkeys and in humans have together indicated that temporal cortex–PFC interactions are involved in visuospatial discrimination learning and recognition (Browning & Gaffan, 2008; Lee et al., 2005; Parker & Gaffan, 1998), with a recent review highlighting the importance of temporal cortex–ventral PFC interactions in supporting visual processing in primates (Eldridge et al., 2021). This neuroimaging evidence has highlighted the crucial contributions of both cortico-thalamocortical and cortico-cortical interactions during the learning of new visuospatial discriminations. However, exactly like the lesion studies, neuroimaging studies also have some disadvantages. For example, this method does not show the complexity of neuron-level signalling or the millisecond responses of neural populations.

For this precision, neurophysiological recording studies are required. Single- and multi-unit recordings targeting both the MD and the PFC during the execution of cognitive tasks in rodents and NHPs have the potential to capture the unique contributions of these structures to cognition. However, such simultaneous recordings have been performed in few studies so far (Perry et al., 2021). One of them (DeNicola et al., 2020) investigated the activity of dorsolateral PFC and lateral MD neurons of NHPs performing a version of the AX-continuous performance test, in which cognitive control is needed to withhold a preponderant response. A rich interaction between both structures was revealed, whereby the timing and response selectivity of PFC neurons suggested a role in context representation, whereas MD neurons coded for the decision and response by the monkey. Interestingly, the pulvinar has also been found to have a role in decision-making, with its firing rate predicting whether or

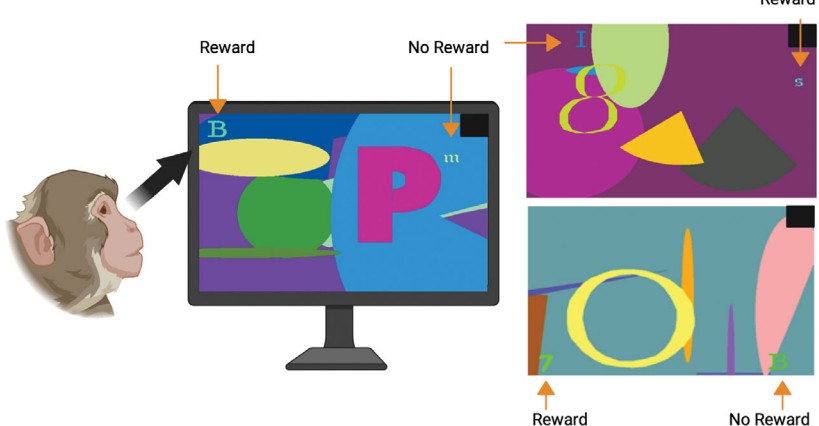

**Figure 2. Visuospatial discrimination task**
For each neurophysiological recording session, the monkeys have to learn, for each trial, which one of two small coloured typographical characters embedded in a unique complex colourful background that includes one large typographical character presented on a touchscreen computer leads to receiving a small amount of fruit smoothie (correct; reward) and which one does not (incorrect; no reward). For a correct choice, the reward is delivered 1.8 s after a screen touch, whereas for an incorrect choice nothing happens for 1.8 s. Up to 30 new, unique visuospatial discriminations are generated for each recording session (depending on the ability of the monkey), and the monkey has to touch the correct typographical character, which they learned by trial and error (i.e. for the first exposure of each discrimination, the monkey has a 50/50 chance of making a correct choice). Each discrimination is presented consecutively and is repeated 16 times within the recording session. Subsequent exposures to each discrimination should result in rapid acquisition of the correct typographical character and a drastic reduction in errors. After the completion of a trial, the screen goes blank, and the monkey receives a 5 s intertrial interval (ITI) for a correct choice or 10 s for an incorrect choice. The black square at the top right-hand corner of each discrimination indicated to a photocell monitoring light intensity that the trial had started and ended.

not a monkey will opt out of a trial depending on its level of confidence associated with receiving a larger reward (Komura et al., 2013).

Given the differential connectivity between the subdivisions of MD and the PFC, a few studies have probed specific neuronal subpopulations to understand the underlying circuits more finely. Anastasiades et al. (2021) showed that the MD contains two populations of thalamocortical projection cells that target either (vasoactive intestinal peptide) VIP$^+$ or (parvalbumin) PV$^+$ cells in the prelimbic PFC, with the former causing an amplification of cortical functional connectivity and the latter a suppression of cortical activity (Mukherjee et al., 2021). These connections proved to be particularly relevant in situations where decisions need to be made in conditions with sparse information (VIP$^+$-targeting) and/or where there is a high level of input noise (PV$^+$-targeting). Likewise, the PFC also contains neuronal subpopulations targeting the MD differentially: the dorsomedial PFC preferentially projects to the more lateral aspects of MD, whereas the ventromedial PFC projects to the more medial MD. Inhibition of the former population during the execution of a five-choice serial reaction-time task in rats decreased premature responses, and the opposite was observed when ventromedial PFC neurons were inhibited (de Kloet et al., 2021). Thus, these findings are revealing the complexities of thalamocortical circuits, in which interacting subpopulations complement each other to control fine behavioural aspects.

## Conclusion

We have proposed previously that the influence of the MD interacting with the cortex contributes as a regulator of cortical functioning when task demands require rapid integration of visual and reward-based information (Mitchell, 2015; Pergola et al., 2018). The inputs from perirhinal cortex and amygdala that are transmitted directly to the PFC and indirectly via the MDmc (see Fig. 1) suggest that this dual pattern of connectivity [cortico-cortical and cortico-thalamocortical (trans-thalamic)] allows the MDmc to help regulate specific neural circuits within the frontal cortex when rapid new learning or updating is required. Likewise, dual patterns of connectivity originate from the midbrain and brainstem, including neuromodulatory transmitters (e.g. noradrenaline and dopamine) and input directly to the PFC and temporal cortex structures and indirectly via the MDmc. These subcortical inputs will also help uniquely to influence and regulate learning and updating, potentially by altering the excitability of the MDmc during these specific aspects of the task.

Interestingly, the pulvinar sends inputs to the perirhinal cortex and the amygdala and has reciprocal connectivity with layer VI of the PFC (e.g. Bos & Benevento, 1975; Elorette et al., 2018; Shipp, 2003; see Fig. 1). The ATN has overlapping connectivity within the PFC and reciprocal connectivity with the subicular complex (not shown here; Perry et al., 2021). This overlapping PFC connectivity highlights the complementary and interdependent influences of the MD, the ATN and the pulvinar for higher cognitive processes carried out by these cortical areas.

In addition, the pulvinar is likely to influence the dorsal stream of visual processing (Milner & Goodale, 1995; Ungerleider & Mishkin, 1982), given its reciprocal connections with the lateral intraparietal cortex (Kagan et al., 2021) and its identified contributions to visuospatial attentional task processing, decision-making and multisensory integration (Froesel et al., 2021; Komura et al., 2013).

For now, we must collect and understand the neurophysiological responses of these cortico-thalamocortical interactions linked to rapid visuospatial discrimination learning, memory and decision-making. To this end, we are currently collecting neurophysiological data from the MD and different frontal cortical areas of NHPs performing a visuospatial discrimination learning task (Chakraborty et al., 2019; Gaffan, 1994; Fig. 2). This task requires continuous rapid learning of visuospatial discriminations and allows us to capture key parts of the learning, memory and decision-making processes. Our hope is that, by the analysis of single- and multi-unit activity and local field potentials, we will gain a deeper understanding of how and when the cortico-MD thalamocortical interactions occur over the course of the visuospatial discrimination task. Others (e.g. Fahy et al., 1993) have found that neurons in the MDmc respond to the previous occurrence of a stimulus during a visual recognition task. Likewise, we have found that both the frontal cortex and the MDmc contain neurons that are tuned to different aspects of the trial, such as when the visual scene is presented, when the monkey makes a choice, and when the reward is delivered (or not). These responses not only appear at crucial time points within a trial, but also seem to evolve over the course of a session, following the learning process of the monkey. Our working proposal is that the MDmc is helping to coordinate neural communication within frontal areas that receive input from specific structures in the temporal cortex, namely the amygdala and the perirhinal cortex, during rapid learning of choice responses to visually relevant (i.e. rewarding) discriminations.

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

## Additional information

### Competing interests

The authors declare no competing interests.

### Author contributions

All authors contributed equally to the drafting and writing of this article. All authors have read and approved the final version of this manuscript and agree to be accountable for all aspects of the work in ensuring that questions related to the accuracy or integrity of any part of the work are appropriately investigated and resolved. All persons designated as authors qualify for authorship, and all those who qualify for authorship are listed.

### Funding

The Thalamus, Cortex and Cognition Lab is supported by a Wellcome Trust Senior Research Fellowship in Basic Biomedical Sciences (110157/Z/15/Z) to A.S.M.

## Acknowledgements

We thank Dr Elsie Premereur for advice and support with adapting the task for neurophysiology recordings; Mr Stuart Mason for training the non-human primates; and Biomedical Services support staff for primate care and husbandry.

## Keywords

anterior thalamus, behaviour, decision making, mediodorsal thalamus, neuroimaging, neurophysiology, prefrontal cortex, pulvinar

## Supporting information

Additional supporting information can be found online in the Supporting Information section at the end of the HTML view of the article. Supporting information files available:

**Peer Review History**

