## [Peer Review History · The Journal of Physiology]

Cortico-thalamocortical interactions for learning, memory, and decision-making

Brook AL Perry, Juan Carlos Mendez, and Anna S Mitchell
DOI: 10.1113/JP282626

Corresponding author(s): Anna Mitchell (anna.mitchell@psy.ox.ac.uk)

The following individual(s) involved in review of this submission have agreed to reveal their identity: Mathieu Wolff (Referee #3)

Review Timeline:

Submission Date:	15-Dec-2021
Editorial Decision:	14-Jan-2022
Revision Received:	10-Jun-2022
Accepted:	30-Jun-2022

Senior Editor: Ian Forsythe

Reviewing Editor: Richard Carson

Transaction Report:

Dear Dr Mitchell,

Re: JP-SR-2021-282626 "Learning about thalamocortical interactions for optimal higher cognitive processes" by Brook AL Perry, Juan Carlos Mendez, and Anna S Mitchell

Thank you for submitting your invited Review-Symposium to The Journal of Physiology. It has been assessed by a Reviewing Editor and by 3 expert referees and I am pleased to tell you that it is considered to be acceptable for publication following satisfactory revision.

The reports are copied at the end of this email. Please address all of the points and incorporate all requested revisions, or explain in your Response to Referees why a change has not been made.

NEW POLICY: In order to improve the transparency of its peer review process The Journal of Physiology publishes online as supporting information the peer review history of all articles accepted for publication. Readers will have access to decision letters, including all Editors' comments and referee reports, for each version of the manuscript and any author responses to peer review comments. Referees can decide whether or not they wish to be named on the peer review history document.

I hope you will find the comments helpful and have no difficulty in revising your manuscript within 4 weeks.

Your revised manuscript should be submitted online using the links in Author Tasks Link Not Available. This link is to the Corresponding Author's own account, if this will cause any problems when submitting the revised version please contact us.

The image files from the previous version are retained on the system. Please ensure you replace or remove any files that have been revised. Your revised submission should include:

- A Word file of the complete text (including figure legends any Tables);
- An Abstract Figure (with legend in the Article file)
- Each figure as a separate, high quality, file;
- A full Response to Referees;
- A copy of the manuscript with the changes highlighted.
- Author profile. A short biography (no more than 100 words for one author or 150 words in total for two authors) and a portrait photograph of the two leading authors on the paper. These should be uploaded, clearly labelled, with the manuscript submission. Any standard image format for the photograph is acceptable, but the resolution should be at least 300 dpi and preferably more.

- A 'Cover Art' file for consideration as the Issue's cover image;
- Appropriate Supporting Information (Video, audio or data set https://jp.msubmit.net/cgi-bin/main.plex?form_type=display_requirements#supp).

To create your 'Response to Referees' copy all the reports, including any comments from the Reviewing Editor into a Word, or similar, file and respond to each point in colour or CAPITALS and upload this when you submit your revision.

I look forward to receiving your revised submission.

If you have any queries please reply to this email and staff will be happy to assist.

Yours sincerely,

Ian D. Forsythe
Deputy Editor-in-Chief
The Journal of Physiology
<https://jp.msubmit.net>
<http://jp.physoc.org>
The Physiological Society
Hodgkin Huxley House
30 Farringdon Lane
London, EC1R 3AW
UK
<http://www.physoc.org>
<http://journals.physoc.org>

REQUIRED ITEMS:

-Please include an Abstract Figure. The Abstract Figure is a piece of artwork designed to give readers an immediate understanding of the Review Article and should summarise the main conclusions. If possible, the image should be easily 'readable' from left to right or top to bottom. It should show the physiological relevance of the Review so readers can assess the importance and content of the article. Abstract Figures should not merely recapitulate other figures in the Review. Please try to keep the diagram as simple as possible and without superfluous information that may distract from the main conclusion of the Review. Abstract Figures must be provided by authors no later than the revised manuscript stage and should be uploaded as a separate file during online submission labelled as File Type 'Abstract Figure'. Please ensure that you include the figure legend in the main article file. All Abstract Figures will be sent to a professional illustrator for redrawing and you may be asked to approve the redrawn figure before your paper is accepted.

-Your MS must include a complete "Additional information section" with the following 4 headings and content:

Competing Interests: A statement regarding competing interests. If there are no competing interests, a statement to this effect must be included. All authors should disclose any conflict of interest in accordance with journal policy.

Author contributions: Each author should take responsibility for a particular section of the study and have contributed to writing the paper. Acquisition of funding, administrative support or the collection of data alone does not justify authorship; these contributions to the study should be listed in the Acknowledgements. Additional information such as 'X and Y have contributed equally to this work' may be added as a footnote on the title page.

It must be stated that all authors approved the final version of the manuscript and that all persons designated as authors qualify for authorship, and all those who qualify for authorship are listed.

Funding: Authors must indicate all sources of funding, including grant numbers. If authors have not received funding, this must be stated.

It is the responsibility of authors funded by RCUK to adhere to their policy regarding funding sources and underlying research material. The policy requires funding information to be included within the acknowledgement section of a paper. Guidance on how to acknowledge funding information is provided by the Research Information Network. The policy also requires all research papers, if applicable, to include a statement on how any underlying research materials, such as data, samples or models, can be accessed. However, the policy does not require that the data must be made open. If there are considered to be good or compelling reasons to protect access to the data, for example commercial confidentiality or legitimate sensitivities around data derived from potentially identifiable human participants, these should be included in the statement.

Acknowledgements: Acknowledgements should be the minimum consistent with courtesy. The wording of acknowledgements of scientific assistance or advice must have been seen and approved by the persons concerned. This section should not include details of funding.

-Author profile(s) must be uploaded via the submission form. Authors should submit a short biography (no more than 100 words for one author or 150 words in total for two authors) and a portrait photograph of the two leading authors on the paper. These should be uploaded, clearly labelled, with the manuscript submission. Any standard image format for the photograph is acceptable, but the resolution should be at least 300 dpi and preferably more. A group photograph of all authors is also acceptable, providing the biography for the whole group does not exceed 150 words.

EDITOR COMMENTS

Reviewing Editor:

Three referees have provided assessments of your submission. As you will discern, the referees differed somewhat in the enthusiasm that they expressed for the piece. It does however appear to me that the manuscript might be revised such that their concerns can be addressed, and their suggestions incorporated. With respect to the specific points raised by Referee

#2, I consider it reasonable to request that your focus is primarily on work that has already been published, rather than research that is currently in progress. It is also apparent that some of the other sections require further refinement, in order that "take-home" messages might be conveyed more effectively. The integration of additional literature concerning research conducted in humans would also be welcome. Please also ensure that all "factual statements" are supported by reference to appropriate works.

Senior Editor:

Thank you for this review. The RE and referees have a number of recommendations to help you revise this Ms. It is important that you think about some figures. JP does not publish review without figures. Technically you have one, but you also need to provide an abstract figure and around 3 diagrams to help explain the key concepts to a broad audience; e.g. background, key methods and key concepts, summary, for example; but other options are possible (take a look at some of our other Reviews). Please also re-write the abstract to contain factual information about the research topic (your reference to the seminar/meeting can go in the acknowledgements) which should come to a clear conclusion in the final sentence. I look forward to reading your revised submission.

REFeree COMMENTS

Referee #1:

This manuscript by Perry, Mendez & Mitchell provides a focussed review of recent work delineating the contributions of thalamocortical interactions to higher cognition. It places particular emphasis on describing the authors' own ongoing efforts - also relayed in a recent Physiology Society Symposium that inspired this submission - to parse the contributions of mediodorsal thalamus (MD) and orbitofrontal cortex (OFC) interactions to the learning and sustained performance of a reward-guided visuospatial decision-making task. The piece leans heavily on cutting edge work in animal models, particularly with non-human primates, where ongoing methodological developments are allowing critical thalamocortical circuits to be characterized with ever-increasing precision. It also speculates about how breakthroughs in this domain may have the potential to generate more targeted therapeutics for neuropsychiatric disorders in the future. As such, the manuscript provides a welcome summary of ongoing basic science work on thalamocortical interactions while also placing this work in the broader context of clinical translation. I have only a small number of minor comments that, if addressed, I feel could further improve the manuscript:

-- As mentioned the review places especially strong emphasis on MD-OFC interactions. Yet, the topic of the review (as per the title) is more broadly focussed on "thalamocortical interactions for higher cognitive processes", and other higher thalamic nuclei besides the MD are only briefly alluded to. The review may benefit from more explicit discussion/speculation of how the MD may work in tandem (or otherwise) with other thalamic nuclei to support the cognitive processes in question. As I'm sure the authors are aware there has been recent excitement about the likely role of the pulvinar in tasks similar to authors' own task of choice (e.g. Jaramillo et al., 2019, Neuron); indeed this nucleus is known to represent maps of the visual field (e.g. Arcaro et al., 2015, J Neurosci) and may very well be implicated in visuospatial aspects of the decision-making task in Figure 1. Might the authors be able to speculate about complementary or distinct role of MD and pulvinar in learning/sustained performance of this task?

-- Similarly, the review as it presently stands places strong emphasis on work in animal models, and there is little discussion of complementary work from the human literature. I understand and am sympathetic to the desire to keep the review relatively focussed, but perhaps there could be some acknowledgement, however brief, of some of the exciting work that is being done with humans in this area (e.g. Kosciessa et al., 2021, Nat Commun; Hwang et al., 2021, eLife) and some of the advantages the 'human model' (direct link to neuropsychology/psychiatry; easily trained on arbitrarily many complex tasks; furnishes data amenable to large-scale modelling; etc.).

-- The title and conclusions refer to "optimal" cognitive processing and performance. In cognitive psychology/neuroscience this term brings some baggage along with it - implications of following some normative or ideal observer strategy, often in line with Bayesian principles. Here, however, the term is not defined and it remains unclear what "optimal performance" on the visuospatial decision-making task should reflect. I suggest dispensing with the use of this term.

-- p.10 typo: "results indicates"

Referee #2:

This an unusual piece to review since it is almost a 'work in progress' declaration by the authors on their budding electrophysiology work. There is a general review around this topic, that is a bit diffuse and sometimes inaccurate, which requires better proof reading and fact checking by the authors (e.g. Anastasiades et al. (2021) showed that the lateral MD ... is factually incorrect). Another puzzling issue is the 'MD connectivity to inferotemporal areas'. In the entire NHP anatomical literature, I am unaware of there being connections of the MD outside of frontal cortical areas, so I have no idea what the authors are referring to here and why it advances their narrative.

It is also clear that the piece was written by multiple people at different times so some findings are mentioned repetitively from different perspectives and sometimes at odds with one another.

Referee #3:

This commissioned paper from Perry et al. provides a brief but useful overview on the role of the mediodorsal thalamus. While the important role of this structure for cognition has been identified decades ago, recent research has attracted a wider attention on its functional role at circuit-level. So the focus is clearly timely and of interest. It would be an excellent start for people not already familiar with an extensive literature. Besides the overview, current lines of research conducted in monkeys are described, allowing to have an idea of the potential dataset that we can expect from this effort. The paper is well written and easy to go through so I only have quite minor comments.

1. On the figure 1 detailing the behavioral procedure, I can see 3, not 2 typographical characters on each screen (e.g., b, m and a large P on the left; l, s and a large 8 and the top right etc), would it be possible to clarify that and perhaps to indicate which ones exactly are the items that the monkey is required to discriminate? For someone not used to that particular task, it is not trivial.

2. The paragraph starting p.4 "Why might dorsal thalamic nuclei be viable targets for treatment in higher cognitive processes?" is interesting but it does not really answer the question. Perhaps it would be possible to add a few further comments to conclude on that or consider an alternative title? The last bit of that section states that the MD also receives subcortical driver input but no information is provided to help the reader to realize if that has an important functional implication? I think Sherman indicates in its 2016 review paper that HO nuclei may contain both FO and HO cells.

3. I found the section titled "A long road to identifying effective treatments" a bit difficult to follow as to me it reads primarily as a review of the main approaches used to examine MD functions with no clear connection with a specific pathology. Perhaps it reads more like a section addressing the system-level implication of the MD as there are quite a few mentions to cortical regions connected to the MD such as the OFC, PFC or the IT? The take-home of that section maybe could be a little clearer and then the title of that section updated to reflect that.

END OF COMMENTS

Confidential Review

15-Dec-2021

10th June 2022

We are very grateful for the three reviewer's time and would like to thank them and our reviewing editor for their helpful and supportive comments and constructive feedback. We have addressed all the comments and revised the manuscript considerable to broaden the overall message – that the MD, anterior thalamus and pulvinar all contribute to influence the cortex during learning, memory, and decision-making.

Thank you for reconsidering our revised manuscript for publication in J Physiology.

Kind Regards
Anna S Mitchell

REQUIRED ITEMS:

-Please include an Abstract Figure. The Abstract Figure is a piece of artwork designed to give readers an immediate understanding of the Review Article and should summarise the main conclusions. If possible, the image should be easily 'readable' from left to right or top to bottom. It should show the physiological relevance of the Review so readers can assess the importance and content of the article. Abstract Figures should not merely recapitulate other figures in the Review. Please try to keep the diagram as simple as possible and without superfluous information that may distract from the main conclusion of the Review. Abstract Figures must be provided by authors no later than the revised manuscript stage and should be uploaded as a separate file during online submission labelled as File Type 'Abstract Figure'. Please ensure that you include the figure legend in the main article file. All Abstract Figures will be sent to a professional illustrator for redrawing and you may be asked to approve the redrawn figure before your paper is accepted.

This is now done and the legend is in the mail file.

-Your MS must include a complete "Additional information section" with the following 4 headings and content:

These four sections have been added to the manuscript.

Competing Interests: A statement regarding competing interests. If there are no competing interests, a statement to this effect must be included. All authors should disclose any conflict of interest in accordance with journal policy.

Author contributions: Each author should take responsibility for a particular section of the study and have contributed to writing the paper. Acquisition of funding, administrative support or the collection of data alone does not justify authorship; these contributions to the study should be listed in the Acknowledgements. Additional information such as 'X and Y have contributed equally to this work' may be added as a

footnote on the title page.

It must be stated that all authors approved the final version of the manuscript and that all persons designated as authors qualify for authorship, and all those who qualify for authorship are listed.

Funding: Authors must indicate all sources of funding, including grant numbers. If authors have not received funding, this must be stated.

It is the responsibility of authors funded by RCUK to adhere to their policy regarding funding sources and underlying research material. The policy requires funding information to be included within the acknowledgement section of a paper. Guidance on how to acknowledge funding information is provided by the Research Information Network. The policy also requires all research papers, if applicable, to include a statement on how any underlying research materials, such as data, samples or models, can be accessed. However, the policy does not require that the data must be made open. If there are considered to be good or compelling reasons to protect access to the data, for example commercial confidentiality or legitimate sensitivities around data derived from potentially identifiable human participants, these should be included in the statement.

Acknowledgements: Acknowledgements should be the minimum consistent with courtesy. The wording of acknowledgements of scientific assistance or advice must have been seen and approved by the persons concerned. This section should not include details of funding.

-Author profile(s) must be uploaded via the submission form. Authors should submit a short biography (no more than 100 words for one author or 150 words in total for two authors) and a portrait photograph of the two leading authors on the paper. These should be uploaded, clearly labelled, with the manuscript submission. Any standard image format for the photograph is acceptable, but the resolution should be at least 300 dpi and preferably more. A group photograph of all authors is also acceptable, providing the biography for the whole group does not exceed 150 words.

Dr Mitchell joined Department of Experimental Psychology at Oxford University in 2004 as a postdoctoral researcher under the tutelage of Professor David Gaffan. Dr Mitchell then established the Thalamus, Cortex and Cognition Lab in 2009 with funding from the Medical Research Council Career Development Award. She is now a Wellcome Trust Senior Research Fellow in Basic Biomedical Sciences and an Associate Professor. Her group studies the neurobiology of learning, memory, and decision-making focusing on the contributions of the dorsal medial thalamus and interconnected structures in health and disease using animal models and humans. Dr Perry joined the Thalamus, Cortex and Cognition Lab in 2018, as a postdoctoral researcher leading on the neurophysiology recordings and analyses in non-

human primates. Dr Mendez joined the Thalamus, Cortex and Cognition Lab in 2020 as a postdoctoral researcher to support the neurophysiology recording analyses and complete neuroimaging studies in non-human primates.

EDITOR COMMENTS

Reviewing Editor:

Three referees have provided assessments of your submission. As you will discern, the referees differed somewhat in the enthusiasm that they expressed for the piece. It does however appear to me that the manuscript might be revised such that their concerns can be addressed, and their suggestions incorporated. With respect to the specific points raised by Referee #2, I consider it reasonable to request that your focus is primarily on work that has already been published, rather than research that is currently in progress. It is also apparent that some of the other sections require further refinement, in order that "take-home" messages might be conveyed more effectively. The integration of additional literature concerning research conducted in humans would also be welcome. Please also ensure that all "factual statements" are supported by reference to appropriate works.

Senior Editor:

Thank you for this review. The RE and referees have a number of recommendations to help you revise this Ms. It is important that you think about some figures. JP does not publish review without figures. Technically you have one, but you also need to provide an abstract figure and around 3 diagrams to help explain the key concepts to a broad audience; e.g. background, key methods and key concepts, summary, for example; but other options are possible (take a look at some of our other Reviews). Please also re-write the abstract to contain factual information about the research topic (your reference to the seminar/meeting can go in the acknowledgements) which should come to a clear conclusion in the final sentence. I look forward to reading your revised submission.

We have now provided an abstract figure and 2 figures in the main text.

REFEREE COMMENTS

Referee #1:

This manuscript by Perry, Mendez & Mitchell provides a focussed review of recent work delineating the contributions of thalamocortical interactions to higher cognition. It places particular emphasis on describing the authors' own ongoing efforts - also relayed in a recent Physiology Society Symposium that inspired this submission - to parse the contributions of mediodorsal thalamus (MD) and orbitofrontal cortex (OFC) interactions to the learning and sustained performance of a reward-guided visuospatial decision-making task. The piece leans heavily on cutting edge work in animal models, particularly with non-human primates, where ongoing methodological developments are allowing critical thalamocortical circuits to be characterized with ever-increasing precision. It also speculates about how breakthroughs in this domain may have the potential to generate more targeted therapeutics for neuropsychiatric disorders in the future. As such, the manuscript provides a welcome summary of ongoing basic science work on thalamocortical interactions while also placing this work in the broader context of clinical translation. I have only a small number of minor comments that, if addressed, I feel could further improve the manuscript:

-- As mentioned the review places especially strong emphasis on MD-OFC interactions. Yet, the topic of the review (as per the title) is more broadly focussed on "thalamocortical interactions for higher cognitive processes", and other higher thalamic nuclei besides the MD are only briefly alluded to. The review may benefit from more explicit discussion/speculation of how the MD may work in tandem (or otherwise) with other thalamic nuclei to support the cognitive processes in question. As I'm sure the authors are aware there has been recent excitement about the likely role of the pulvinar in tasks similar to authors' own task of choice (e.g. Jaramillo et al., 2019, Neuron); indeed this nucleus is known to represent maps of the visual field (e.g. Arcaro et al., 2015, J Neurosci) and may very well be implicated in visuospatial aspects of the decision-making task in Figure 1. Might the authors be able to speculate about complementary or distinct role of MD and pulvinar in learning/sustained performance of this task?

We have adapted the text throughout to include studies on the pulvinar including these references (and others too). Further, in order to slightly broaden the manuscript, we have added in some details about the pulvinar and anterior thalamus (ATN) connections (and included these in a new figure), to highlight how these connections in the PFC overlap with MD connections and propose the interdependent contributions of these three dorsal thalamic nuclei for influencing cognitive processes. We have speculated in the conclusion that the MD (and the ATN) influences the frontal-temporal communication while the pulvinar influences the frontal-parietal communication.

-- Similarly, the review as it presently stands places strong emphasis on work in animal models, and there is little discussion of complementary work from the human literature. I understand and am sympathetic to the desire to keep the review relatively focussed, but perhaps there could be some acknowledgement, however brief, of some of the exciting work that is being done with humans in this area (e.g. Kosciessa et al., 2021, Nat

Commun; Hwang et al., 2021, eLife) and some of the advantages the 'human model' (direct link to neuropsychology/psychiatry; easily trained on arbitrarily many complex tasks; furnishes data amenable to large-scale modelling; etc.).

We have included these details now.

-- The title and conclusions refer to "optimal" cognitive processing and performance. In cognitive psychology/neuroscience this term brings some baggage along with it - implications of following some normative or ideal observer strategy, often in line with Bayesian principles. Here, however, the term is not defined and it remains unclear what "optimal performance" on the visuospatial decision-making task should reflect. I suggest dispensing with the use of this term.

We have now deleted this and adjusted the title to say Cortico-thalamocortical interactions for learning, memory, and decision-making

-- p.10 typo: "results indicates" Sorted

Referee #2:

This is an unusual piece to review since it is almost a 'work in progress' declaration by the authors on their budding electrophysiology work. There is a general review around this topic, that is a bit diffuse and sometimes inaccurate, which requires better proof reading and fact checking by the authors (e.g. Anastasiades et al. (2021) showed that the lateral MD ... is factually incorrect). Another puzzling issue is the 'MD connectivity to inferotemporal areas'. In the entire NHP anatomical literature, I am unaware of there being connections of the MD outside of frontal cortical areas, so I have no idea what the authors are referring to here and why it advances their narrative.

We have rechecked Figure 2 of Anastasiades et al. (2021) *Neuron*, while the GFP labelling appears to indicate the target of the experiments as the lateral MD, we have now deleted lateral and instead now refer in our manuscript to the MD.

In the non-human primate, there is connectivity from the temporal lobes (perirhinal cortex and entorhinal cortex) to the MD, it is unidirectional (e.g. Russchen, Price and Amaral, 1987, *J Comp Neurol*; Saunders, Mishkin, Aggleton, 2005; also reviewed by Aggleton and Brown, 1999, *Beh Brain Sci*).

We have adjusted the text to be more specific about the temporal lobes structures and included specific references.

We have also included another figure to help capture this connectivity along with highlighting the overlap of pulvinar and anterior thalamic connectivity in the frontal cortex and temporal cortex.

It is also clear that the piece was written by multiple people at different times so some findings are mentioned repetitively from different perspectives and sometimes at odds with one another.

We have adjusted the text to remove the repetition.

Referee #3:

This commissioned paper from Perry et al. provides a brief but useful overview on the role of the mediodorsal thalamus. While the important role of this structure for cognition has been identified decades ago, recent research has attracted a wider attention on its functional role at circuit-level. So the focus is clearly timely and of interest. It would be an excellent start for people not already familiar with an extensive literature. Besides the overview, current lines of research conducted in monkeys are described, allowing to have an idea of the potential dataset that we can expect from this effort. The paper is well written and easy to go through so I only have quite minor comments.

1. On the figure 1 detailing the behavioral procedure, I can see 3, not 2 typographical characters on each screen (e.g., b, m and a large P on the left; l, s and a large 8 and the top right etc), would it be possible to clarify that and perhaps to indicate which ones exactly are the items that the monkey is required to discriminate? For someone not used to that particular task, it is not trivial.

This figure has been adjusted to indicate the target typographical characters – it is now Figure 2. We have also clarified these details in the figure legend.

2. The paragraph starting p.4 "Why might dorsal thalamic nuclei be viable targets for treatment in higher cognitive processes?" is interesting but it does not really answer the question. Perhaps it would be possible to add a few further comments to conclude on that or consider an alternative title? The last bit of that section states that the MD also receives subcortical driver input but no information is provided to help the reader to realize if that has an important functional implication? I think Sherman indicates in its 2016 review paper that HO nuclei may contain both FO and HO cells.

We have provided further details about first order and higher order cells and referred to the Sherman 2016 reference.

3. I found the section titled "A long road to identifying effective treatments" a bit difficult

to follow as to me it reads primarily as a review of the main approaches used to examine MD functions with no clear connection with a specific pathology. Perhaps it reads more like a section addressing the system-level implication of the MD as there are quite a few mentions to cortical regions connected to the MD such as the OFC, PFC or the IT? The take-home of that section maybe could be a little clearer and then the title of that section updated to reflect that.

We have reduced this section and deleted the details that were not relevant to our message.

END OF COMMENTS

Dear Dr Mitchell,

Re: JP-SR-2022-282626R1 "Cortico-thalamocortical interactions for learning, memory, and decision-making" by Brook AL Perry
Juan Carlos Mendez
Anna S Mitchell

I am pleased to tell you that your Symposium Review article has been accepted for publication in The Journal of Physiology, subject to any modifications to the text that may be required by the Journal Office to conform to House rules.

NEW POLICY: In order to improve the transparency of its peer review process The Journal of Physiology publishes online as supporting information the peer review history of all articles accepted for publication. Readers will have access to decision letters, including all Editors' comments and referee reports, for each version of the manuscript and any author responses to peer review comments. Referees can decide whether or not they wish to be named on the peer review history document.

The last Word version of the paper submitted will be used by the Production Editors to prepare your proof. When this is ready you will receive an email containing a link to Wiley's Online Proofing System. The proof should be checked and corrected as quickly as possible.

All queries at proof stage should be sent to tjp@wiley.com

The accepted version of the manuscript is the version that will be published online until the copy edited and typeset version is available. Authors should note that it is too late at this point to offer corrections prior to proofing. Major corrections at proof stage, such as changes to figures, will be referred to the Reviewing Editor for approval before they can be incorporated. Only minor changes, such as to style and consistency, should be made a proof stage. Changes that need to be made after proof stage will usually require a formal correction notice.

Are you on Twitter? Once your paper is online, why not share your achievement with your followers. Please tag The Journal (@jphysiol) in any tweets and we will share your accepted paper with our 22,000+ followers!

Yours sincerely,

Ian D. Forsythe
Deputy Editor-in-Chief
The Journal of Physiology
<https://jp.msubmit.net>
<http://jp.physoc.org>
The Physiological Society
Hodgkin Huxley House
30 Farringdon Lane
London, EC1R 3AW
UK
<http://www.physoc.org>
<http://journals.physoc.org>

Comments:

Reviewing Editor:

The authors are to be commended in respect of the positive changes that were made to address concerns raised by the referees.

Senior Editor:

Many thanks for an interesting Review - I look forward to seeing it in print.

REFEREE COMMENTS:

Referee #1:

I thank the authors for engaging well with my suggestions and incorporating them into the revised manuscript. I have no further comments.

Referee #3:

My comments have been addressed.

* IMPORTANT NOTICE ABOUT OPEN ACCESS *

To assist authors whose funding agencies mandate public access to published research findings sooner than 12 months after publication The Journal of Physiology allows authors to pay an open access (OA) fee to have their papers made freely available immediately on publication.

You will receive an email from Wiley with details on how to register or log-in to Wiley Authors Services where you will be able to place an OnlineOpen order.

You can check if your funder or institution has a Wiley Open Access Account here <https://authorservices.wiley.com/author-resources/Journal-Authors/licensing-and-open-access/open-access/author-compliance-tool.html>

Your article will be made Open Access upon publication, or as soon as payment is received.

If you wish to put your paper on an OA website such as PMC or UKPMC or your institutional repository within 12 months of publication you must pay the open access fee, which covers the cost of publication.

OnlineOpen articles are deposited in PubMed Central (PMC) and PMC mirror sites. Authors of OnlineOpen articles are permitted to post the final, published PDF of their article on a website, institutional repository, or other free public server, immediately on publication.

Note to NIH-funded authors: The Journal of Physiology is published on PMC 12 months after publication, NIH-funded authors DO NOT NEED to pay to publish and DO NOT NEED to post their accepted papers on PMC.

1st Confidential Review

10-Jun-2022